# Severe Neutropenia and Agranulocytosis Related to Antithyroid Drugs: A Study of 30 Cases Managed in A Single Reference Center

**DOI:** 10.3390/medicines7030015

**Published:** 2020-03-19

**Authors:** Emmanuel Andrès, Noel Lorenzo-Villalba, Rachel Mourot-Cottet, Frédéric Maloisel, Martine Tebacher, Jacques-Eric Gottenberg, Bernard Goichot, Raoul Herbrecht, Abrar-Ahmad Zulfiqar

**Affiliations:** 1Departments of Internal Medicine, Strasbourg University Hospitals, 67000 Strasbourg, France; noellorenzo@gmail.com (N.L.-V.); rachel.mourot-cottet@chru-strasbourg.fr (R.M.-C.); bernard.goichot@chru-strasbourg.fr (B.G.); abzulfiqar@gmail.com (A.-A.Z.); 2Onco-Hematology, Strasbourg University Hospitals, 67000 Strasbourg, France; frederic.maloisel@stanne.fr (F.M.); raoul.herbrecht@chru-strasbourg.fr (R.H.); 3Regional Pharmacovigilance Centre of Alsace, 67000 Strasbourg, France; martine.tebacher@chru-strasbourg.fr; 4Rheumatology, Strasbourg University Hospitals, 67000 Strasbourg, France; jacques-eric.gottenberg@chru-strasbourg.fr

**Keywords:** neutropenia, agranulocytosis, antithyroid drug, carbimazole, propylthiouracil, diagnosis, fever, infection, hematopoietic growth factor, granulocyte colony-stimulating factor

## Abstract

**Background:** The most important series devoted to antithyroid drug-induced severe neutropenia and agranulocytosis are Japanese studies, almost specifically in relation to the intake of methimazole. The clinical data of 30 Caucasian patients followed up for antithyroid drug-induced neutropenia at a third-level hospital are reported. **Methods:** The data of 30 patients with idiosyncratic antithyroid drug-induced neutropenia and agranulocytosis from a cohort study on drug-induced neutropenia and agranulocytosis conducted at the University Hospital of Strasbourg (France) were retrospectively reviewed. **Results:** The mean patient age was 61.7 years old (range: 20–87), and the gender ratio (F/M) was 4. Several comorbidities were reported in 23 patients (76.7%), with the mean Charlson comorbidity index of 1. The causative drugs were carbimazole and benzylthiouracil, in 28 (93.3%) and 2 cases, respectively, prescribed primarily for multi-hetero-nodular goiter or thyroid nodule to 18 patients (60%). Sore throat and acute tonsillitis (40%), isolated fever (20%), septicemia (13.3%), documented pneumonia (6.7%), and septic shock (6.7%) were the main clinical features upon admission. The mean neutrophil count at nadir was 0.02 and 0 × 10^9^/L (range: 0–0.3). Regarding the patients’ hospital course: 13 cases (43.3%) worsened during hospitalization, severe sepsis was found in 26.7%, systemic inflammatory response syndrome—in 13.3%, and septic shock—in 3.3% of the cases, respectively. Broad-spectrum antibiotics were indicated for all the patients, and 21 (73.3%) of them received hematopoietic growth factors. Hematological recovery (neutrophil count ≥ 1.5 × 10^9^/L) was seen at 8.3 days (range: 2–24), but faster in those receiving hematopoietic growth factors (4.9 days, *p* = 0.046). Two patients died during hospitalization, and the rest had a favorable clinical outcome. **Conclusions:** Antithyroid drug-induced neutropenia represents a serious complication resulting from the rates of severe infections especially in those cases severe neutropenia. In this setting, an established procedure for the management of patients seems useful or even indispensable in view of potential mortality.

## 1. Introduction

Synthetic antithyroid drugs (ATDs), prescribed for almost 50 years, include several molecules—derivatives of thioureas. They include carbimazole, methimazole, thiamazole (active form of carbimazole), benzylthiouracil, and propylthiouracil [1]. These molecules are used in the majority of hyperthyroid patients, either over a long term (e.g., for Basedow’s or Graves’ disease) or before thyroid surgery or prescription of radioactive iodine [1]. In practice, ATDs have been associated with idiosyncratic neutropenia and agranulocytosis, defined as the absolute blood neutrophil count ≤ 1.5 × 10^9^/L and ≤ 0.5 × 10^9^/L with fever or ≤ 0.1 × 10^9^/L, respectively [2]. In this setting, the most important series devoted to ATD-induced severe neutropenia and agranulocytosis are Japanese studies, almost specifically in relation to the intake of methimazole [2,3,4]. To our knowledge, there is only one study featuring use of carbimazole, the most widely used molecule in Europe and the USA, in Caucasian patients [5].

The present article shows the results of the follow-up of 30 patients with ATD-related (mostly carbimazole) neutropenia and agranulocytosis at a university hospital.

## 2. Methods

### 2.1. Patient Selection

All non-chemotherapy-related neutropenia and agranulocytosis cases identified in patients hospitalized at the University Hospitals of Strasbourg (Hôpitaux Universitaires de Strasbourg), France, were included into a register since the 1990′s (partial data published in [6]). Patients were initially hospitalized at the Internal Medicine, Hematology, Endocrinology, Oncology, Geriatric, Rheumatology, or General Surgery departments. During the hospitalization, all these patients were managed and followed up at the Internal Medicine or Hematology departments of our institution. A protocol aiming at optimizing the management of idiosyncratic drug-induced neutropenia and agranulocytosis was established at the university hospital [6].

### 2.2. Inclusion Criteria

The inclusion criteria included the neutrophil count ≤ 0.5 × 10^9^/L, fever, and any kind or degree of clinical infection, as well as the standardized criteria defined by Benichou et al. [7,8]. Patients with a previous history of congenital neutropenia or immune neutropenia, hematological diseases, recent viral or bacterial infections, or those previously receiving chemotherapy, radiotherapy, or immunotherapy were excluded. Besides, all the patients fulfilled the criteria “likely causative” or “very likely causative” according to the French causality assessment method of adverse drug reactions [9]. All the patients were registered at the Pharmacovigilance Centre of Alsace (Centre regional de Pharmacovigilance d’Alsace), France. Neutropenia was classified into several categories according to its severity, as proposed in Oncology as part of the treatment of tumors by chemotherapy (https://www.cancer.gov/). This classification proposed by the National Cancer Institute includes four categories ranging from Grade 1 to 4. Grade 1: absolute neutrophil count from 1.5 to 1 × 10^9^/L, Grade 2—from 1 to 0.5 × 10^9^/L, Grade 3—from 0.5 to 0.1 × 10^9^/L, and Grade 4 ≤ 0.1 × 10^9^/L. Severe neutropenia, also known as agranulocytosis, is biologically characterized by a severe decrease or an absolute lack of circulating granulocytes, resulting in an absolute neutrophil count of ≤ 0.5 × 10^9^/L with fever or absolute neutrophil count ≤ 0.1 × 10^9^/L (Table 1). In the present study, we only included neutropenia and agranulocytosis related to synthetic ATDs. In France, these agents include carbimazole (Néomercazole^®^), methimazole (Thyrozol^®^), benzylthiouracil (Basdène^®^), and propylthiouracil (Proracyl^®^, or PTU). The active form of carbimazole, thiamazole, is not available in France.

### 2.3. Objective, Method, and Collected Data

This study was aimed at describing the clinical features, management, and outcome of all ATD-related neutropenia and agranulocytosis cases, with a focus on carbimazole, the main ATD used in France and Europe. The data were obtained from the medical records. The medical record of every patient was assessed by two members of the monitoring committee, and the following variables were recorded (when available): age, sex, reason for hospitalization, associated comorbidities, and Charlson comorbidity index. We collected information on the underlying thyroid disease and the synthetic antithyroid drug used (dose, administration route, start and withdrawal dates). The clinical features upon diagnosis and the disease course were evaluated. Complete white blood cell counts, neutrophil counts, hemoglobin and platelet counts (in absolute values) were collected as the bone marrow aspects. The delay to reach a neutrophil count > 1.5 × 10^9^/L (hematological recovery), length of hospitalization, outcome, transfer to intensive care unit, and mortality were collected with special regard to the use of hematopoietic growth factors (HGFs).

### 2.4. Statistical Analysis

The data were expressed as the mean and the standard deviation (SD). The Mann–Whitney test and the Student’s *t*-test were used for paired data. Quantitative variables distribution was graphically assessed through the Shapiro–Wilk test. The nonparametric Mann–Whitney test was used for in between-group comparison of quantitative variables. The qualitative variables were presented in absolute numbers and percentages and were analyzed using the Pearson’s chi-squared or the Fisher’s exact test. To assess the factors associated with mortality and/or intensive care unit admission, a univariate analysis was performed. The results were considered statistically significant with the *p*-value < 0.05. We used the R software (version 3.2.2) for the statistical analyses.

### 2.5. Administrative Data

The data collected were registered in the French National Database of drug side effects (Banque Nationale du Réseau des Centres Régionaux de Pharmacovigilance) (https://www.rfcrpv.fr/) and were subject to the French National Data Protection Act (Commission Nationale Informatique et Liberté) (CNIL) (https://www.cnil.fr/professionnel). The local ethics committee approved the present study. Ethical approval code: CEFM_1996_3, Date of approval: 4 March 1996. Written informed consent for publication was obtained from the participating patients.

## 3. Results

A total of 30 cases of neutropenia related to ATDs were registered in the period from January 1990 to January 2020 (30 years). In the same study period, 177 patients were hospitalized for neutropenia related to other drugs (e.g., antibiotics, nonsteroidal anti-inflammatory agents, clozapine, and ticlopidine). One female patient had two episodes of drug-induced agranulocytosis three years apart, the first one under carbimazole (daily dose of 40 mg) and the second one under benzylthiouracil treatment (daily dose of 100 mg), respectively.

### 3.1. Patient Characteristics

All the patients had a Caucasian background. In the sample, the mean and the median age was 61.7 and 66 years, respectively; 7 patients (23.3%) were aged < 50 years, and 14 (4.7%)—< 65 years. Only 30 percent (9) of the patients were older than 75 years. There was a female predominance (sex ratio 4). In 23 patients (76.7%), the coexistence of multiple medical conditions was observed, consisting mainly of arterial hypertension (n = 12, 40%); atrial fibrillation (n = 10, 33.3%); chronic renal failure (n = 4, 13.3%); cardiac disorders (myocardial infarction, chronic heart failure) (n = 3, 10%), arteritis of lower limbs (n = 3, 10%); diabetes mellitus (n = 3; 10%); stroke (n = 2, 6.7%); and rheumatoid arthritis (n = 2, 6.7%). Moreover, 2 patients (6.7%) were followed up for a non-progressing and treated Waldenström’s disease. The mean Charlson comorbidity index was 1 (range: 0–4).

### 3.2. Causative Drugs and Underlying Thyroid Disorders

In all but two cases (6.7%), a synthetic ATD was found to be “causative” or “likely causative”. In these two cases, other “likely causative” drugs (flecainide, Nureflex) were related to the biological picture. Carbimazole in 28 patients (93.3%) and benzylthiouracil in 2 cases were identified as the causative drugs. These drugs were discontinued during the first 48 h after hospital admission in 67.9% of the patients. The mean and the median daily doses of carbimazole and benzylthiouracil were 48 and 40 mg (40–100) and 150 and 100 mg, respectively. The mean and the median treatment duration for carbimazole was 42.2 days and 35 days (18–120), respectively. Regarding benzylthiouracil, the patient receiving 150 mg was treated for 3 days, and the other one (100 mg)— 8 days.

The underlying thyroid disorders were: multi-hetero-nodular goiter or thyroid nodule in 18 patients (60%); administration of Cordarone for atrial fibrillation in 9 patients (30%); and Basedow‘s (Graves’) disease in 3 cases (10%). Twenty-two patients (73.3%) were under treatment with at least one drug (mean number of drugs: 5). In eight patients (26.7%), no other medication except the causative drug was identified within the 10 days prior to the detection of neutropenia.

### 3.3. Clinical Manifestations

Fever of unknown origin was the clinical sign leading to neutropenia detection in 15 patients (50%). Neutropenia was found in 13 patients (43.3%) in the absence of any symptoms as part of routine monitoring of their blood counts. Only 2 patients (6.7%) had a documented infection with pneumonia and septic shock. Sore throat and acute tonsillitis in 12 patients (40%) and isolated fever in 6 patients (20%) were the principal clinical findings on admission. Table 1 summarizes data for all the patients. Three patients (10%) remained asymptomatic throughout their hospitalization. In one case (female, 77 years old, under carbimazole), there were signs of drug toxidermia accompanying necrotizing angina. In case of Grade 3 neutropenia, 90% of the patients (n = 10) only had isolated fever or acute tonsillitis. The clinical presentation did not differ between the patients aged < 75 years and ≥ 75 years (non-detailed data; *p* > 0.05). In 6 patients (20%), microbiology results were contributive: gram-negative bacilli (n = 4), *Staphylococcus aureus* and *Streptococcus pneumoniae*.

During hospitalization, clinical worsening was noted in 13 patients (43.3%) ranging from severe sepsis (n = 8, 26.7%), systemic inflammatory response syndrome (SIRS) (n = 4, 13.3%) to septic shock (n = 1, 3.3%). The worsening of the clinical condition was initially observed in 62.5% of patients over 75 years of age compared to 36.4% of other patients. Two patients (6.6%) had to be admitted to intensive care due to visceral failures related to septic shock.

### 3.4. Hematological Data

The mean and the median absolute neutrophil counts were 0.08 and 0.02 × 10^9^/L (range: 0–0.5) upon admission. Twenty patients (66.7%) had the absolute neutrophil level ≤ 0.1 × 10^9^/L (Grade 4 neutropenia). All the remaining patients had Grade 3 neutropenia. In this setting, 90% of the patients presented with a mild clinical picture of isolated fever or acute tonsillitis. Concerning neutrophil nadir, the mean and the median neutrophil counts were 0.02 and 0 × 10^9^/L (range: 0–0.3), respectively. Ninety percent of the patients (n = 27) had the neutrophil level ≤ 0.1 × 10^9^/L (Grade 4). Isolated neutropenia without any other blood abnormalities was found in 26 patients (86.7%). The mean and the median hemoglobin levels were 113 and 110 g/L, respectively. Anemia (Hb < 120 g/L) was found in 16 patients (53.3%). Regarding the platelet counts, the mean value was 281, and the median level was 278 × 10^9^. Thrombocytopenia was not detected in any patient. The bone marrow study was available in 13 cases: primarily detected a partial disappearance or even aplasia in 7 patients (53.8%) and myeloid hypocellularity in 4 other patients (30.8%). The bone marrow was described as normal in 2 cases (15.4%).

### 3.5. Time to Reach Hematological Recovery and Response to Hematopoietic Growth Factors

The hematological recovery (time from initial neutropenia detection to absolute neutrophil count ≥ 1.5 × 10^9^/L) was 8.3 and 6 days (the mean and the median), respectively. As for achieving the neutrophil count ≥ 0.5 × 10^9^/L, these values were 8.1 and 6 days.

HGFs were administered in 21 cases (73.3%), especially in those presenting with the neutrophil count ≤ 0.1 × 10^9^/L, severe clinical infections, renal impairment, or in patients older than 75 years. This medication was administered in a daily fixed dose of 300 μg. Only one patient received granulocyte–macrophage colony-stimulating factors (GM-CSF). Of note, HGFs were started immediately after diagnosis, except in 1 patient with isolated fever. The duration of this treatment administration was 5.9 and 5 days (the mean and the median). In patients receiving this treatment, the mean time to hematological recovery dropped to 4.9 days, from 12.2 to 7.2 days (*p* = 0.046).

### 3.6. Management, Duration of Hospitalization, and Outcome

Because of the severity of neutropenia, all the patients were hospitalized. In 19 patients (63.3%), the incriminated drugs were withdrawn within the first 48 h. In case of fever (> 38 °C) or any signs of infection, broad-spectrum parenteral antibiotherapy, mainly an association of cephalosporins and aminoglycosides, was initiated if no ß-lactam allergy was present. Subsequently, this treatment was adapted to the microbiology results, and other antibiotics, such as fluoroquinolone, glycopeptides and penems, were used. Antibiotics were administered for 11.7 and 12 days (the mean and the median), and the hospital stay length was 17.8 and 12 days (the mean and the median), respectively. Antibiotics and hospital stay length were not influenced by the use of G-CSF therapy (data not shown; all *p* > 0.005).

During the hospitalization, 2 patients (6.6%) were admitted to intensive care due to multiple organ failure related to septic shock. The outcome was favorable in 28 patients (93.3%), whereas 2 subjects (6.7%) died of either uncontrolled septicemia due to *Escherichia coli* related to an initial pyelonephritis in a 77-year-old female and a *Pseudomonas aeruginosa* and *Morganella morganii* septic shock related to an initial inguinal abscess in a 73-year-old female. The first patient was under benzylthiouracil (daily dose of 150 mg) for cardiothyreosis related to Cordarone intake; the second one was also under carbimazole (daily dose of 40 mg) for hyperthyroidism related to Cordarone intake. Of note, HGFs were used in these two patients.

## 4. Discussion

This is the first study of well-documented neutropenia related to ATDs (primarily carbimazole) focusing on Grade 3, Grade 4 neutropenia and agranulocytosis and reporting a detailed clinical picture for each patient, and managed using the same procedure. To date, the most important series devoted to ATD-induced neutropenia are primarily local or regional epidemiological studies [3,4,5,10]. They included neutropenia of varying severity, with rare cases of documented agranulocytosis. Moreover, these studies mainly, if not exclusively, included cases of agranulocytosis related to methimazole. In this setting, one of the most important studies (monocentric Japanese study, conducted from 1975 to 2001) included 93 cases of neutropenia related to methimazole in 109 cases [4]. To our knowledge, the only exception is an English study (1963-2003) of the Great Britain’s National Pharmacovigilance Agency which reports 94 cases of carbimazole- (88.7%) and 12 cases of propylthiouracil-induced agranulocytosis [2,5].

Our patients met the definition of idiosyncratic drug-induced neutropenia and agranulocytosis [7,11] (Table 1). All cases showed established neutropenia: mean neutrophil count of 0.02 × 10^9^/L at the neutrophil nadir; and 90% had the level ≤ 0.1 × 10^9^/L (Grade 4 neutropenia) at nadir. In all cases except for the two patients who died, the principal diagnostic criterion was fulfilled with a full hematological recovery once the causative drug was discontinued [8]. Considering the severity of the complications, the patients were not exposed to the considered causative drug, which is theoretically considered the reference method [8]. A synthetic ATD was identified as “causative” or “likely causative” in 93.3% of cases. These drugs, carbimazole (n = 28) and propylthiouracil (n = 2), were prescribed and administered according to the usual conditions of use (indications, doses, durations of therapy, monitoring of the blood count).

In our study, the clinical features differ from those described in other series of ATD-induced neutropenia [3,4,5], and especially in series of idiosyncratic agranulocytosis including all drug classes [11,12]. In fact, transient Grade 1 neutropenia (neutrophil count from 1.5 to 1 × 10^9^/L) is relatively common with ATDs and is often included in other series [2]. In this setting, ATD-induced neutropenia may often be asymptomatic [2,13]. This Grade 1 neutropenia discovery is linked to the routine follow-up of a patient under antithyroid treatment as recommended by several authors and endocrinology societies [2,14]. In this setting, life-threatening manifestations have been reported in more than 2/3 of the cases [11,12,15]. In our sample, the principal clinical manifestations were sore throat and acute tonsillitis (40%), isolated fever (20%). Septicemia was reported in only 13.3% of the patients. Forty three percent of the patients developed infections (of any kind) during hospitalization, but only 6.6% (n = 2) of the patients were transferred to the intensive care unit. The severity of the clinical features could be explained by the degree of neutropenia and also by the fact that only hospitalized patients were included.

The mortality rate was 6.6% in relation to Grade 3 and Grande 4 neutropenia and agranulocytosis, with one case of uncontrolled septicemia complicated by pyelonephritis, and one other case of septic shock in two old females. In this setting, the literature related to ATD-induced severe neutropenia and agranulocytosis for the last 30 years shows a progressive decrease in mortality with each decade. A Swedish study conducted in 1966–1975 reported 5 deaths among 29 cases of agranulocytosis induced by ATDs (17%); the risk appeared similar for carbimazole, methimazole, and propylthiouracil [16]. The Cooper’s study (1953–1981) and the International Agranulocytosis and Aplastic Anaemia Study (IAAAS; 1980-1986) each found the mortality rate of about 2% [17]. Finally, the Pearce’s retrospective study from 1963 to 2003 showed the mortality rate of 18% [5]. No significant difference in mortality is found between patients taking carbimazole and those exposed to propylthiouracil. However, mortality seems to be more pronounced in individuals aged 65: 13.8% versus 1.2% (RR: 12.9, 95% CI: 1.45–114.9). The improvement in knowledge of the pathophysiology and the optimization of treatment, particularly with regard to the use of antibiotic combinations, are probably at the root of this significant reduction in mortality [6,12]. For example, there were no deaths in 109 cases of agranulocytosis related to ATDs managed at a Japanese endocrinology reference center where this pathology is known (numerous publications from this center), diagnosed early and treated in a codified manner, with the systematic use of HGFs (mainly G-CSFs) in particular [18]. One other explanation of the low reported mortality may be an early diagnosis of neutropenia as part of blood count monitoring as recommended by international thyroid societies [2,14]. The mortality of 6.6% represents one of the lowest rates reported in the literature even though our patients were relatively old and frail and with several comorbidities in 76.7% of cases. One explanation may be the low mean Charlson comorbidity index of 1 in our sample. It should be noted that this low mortality was observed despite Grade 4 neutropenia in 90% of the patients during the course of the disease. Renal failure and neutrophil count ≤ 0.1 × 10^9^/L were found to be poor prognostic factors leading to infections in the study carried out by Julia et al. [15]. Age (75 years), infection severity, and comorbidities (especially renal failure) were shown to have a negative prognostic impact on neutropenia according to Maloisel et al. [19].

In 94.4% of patients, the outcome is favorable, and they all benefit from the established protocol of care (for details see [6,7]). This protocol of care may explain the good clinical and biological results observed in our study in spite of the neutropenia severity and the clinical features. In the context of hyperthyroidism, stopping an ATD is not a major endocrine problem in the short term, since the inhibitory effect on the thyroid function is prolonged [1,2]. In patients with rhythmic and/or ischemic heart disease or with life-threatening hyperthyroidism, substitution of an ATD with another antithyroid molecule may be necessary (a procedure not accepted by all experts). However, the existence of cross-reactions between ATDs has been documented, particularly between carbimazole and propylthiouracil (around 15%) [16,20,21]. In this setting, particularly in case of symptomatic hyperthyroidism, patients should have radioactive iodine treatment or surgery, often with Lugol’s solution pretreatment [22]. Sepsis in the setting of ATD-induced neutropenia requires hospitalization and immediate antibiotherapy [6,7,12]. Hospitalization should also be considered in asymptomatic patients at high risk of infection [7,11]. We strongly recommend the use of broad-spectrum antibiotics and HGFs when managing sepsis in these patients as prognosis could be improved [6,7,12].

Hematological improvement was rapidly observed in the HGF group: -4.9 days (*p* = 0.046), but no improvement concerning antibiotherapy and hospital stay length. Our results are similar to those described in other studies [12], but are not consistent with those from our previous study dedicated specifically to ATDs (n = 20) [23]. Statistically significant differences in favor of the use of HGFs have been observed for hematological recovery times (6.8 versus 11.6 days; *p* = 0.046), and also for hospitalization duration (7.3 versus 13 days; *p* = 0.038). As no fatal cases were observed, the benefit on mortality could not be studied. To date, there are two other published clinical studies specifically dedicated to ATD-induced neutropenia and agranulocytosis treated with HGFs [18,24]. The first study is a prospective randomized Japanese study involving 24 patients with documented agranulocytosis related to ATDs [24]. In the Fukata’s study, there was no significant reduction in the average duration of agranulocytosis. However, certain limitations must be pointed out that make it difficult to interpret the results of this study. First, the sample size was small, and the dosage of G-CSFs (< 200 µg/day) was far from what is currently considered effective [11]. In the second study, by Tajiri et al., in of a total of 109 patients with agranulocytosis, use of HGFs resulted in a duration of agranulocytosis shorter by two days (G-CSFs were used in 34 cases) [18]. In this study as well, since no fatal cases were observed, the benefit on mortality could not be studied. It should be noted that in this work, the matching criteria were met, but the daily dosage of G-CSFs used (75 µg) was very low compared to the currently recommended dosage [11].

The medical literature recommends monitoring the neutrophil count when certain drugs, such as clozapine, ticlopidine and ATDs, are prescribed [14,25]. The results of the present study seem to support this recommendation. The blood neutrophil count at every medical visit when prescribing ATDs has recently demonstrated to correctly diagnose 64% and 94% of patients with agranulocytosis without or with minimum infection symptoms when prescribing ATDs [14,26]. However, the lack of impact on mortality and morbidity leads to a debate regarding blood count monitoring [7,8]. This might explain the absence of blood cell count monitoring in patients under treatment with certain groups of medication, such as ticlopidine or ATDs [7,11]. At this level, it is imperative to highlight the importance of patient education in preventing the most serious accidents.

Our study has some limitations as the data were obtained from patients over 30 years of age, which makes the management heterogeneous (e.g., antibiotics, HGFs). However, this is the first study mainly evaluating carbimazole-induced neutropenia in Caucasian patients. It is important to highlight the presence of Grade 4 neutropenia in 90% of the patients during the biological course. The patients were managed by the physicians experienced in this field and in a single center, which makes it difficult to extrapolate these results. The sample size is small, and the statistical analysis could not take into consideration any confounding factors. A large-scale study including different centers could be of help to confirm our findings.

## 5. Conclusions

Idiosyncratic ATD-induced neutropenia and agranulocytosis are serious medical conditions as severe sepsis was found in half of the patients and the mortality rate was 6.7%. Updated management including immediate antibiotherapy and HGFs in frail patients may reduce infection-related mortality. A specific protocol of care seems to be indispensable in view of the mortality rate. Although ATD-induced neutropenia is a known adverse drug reaction, many questions remain unanswered. The studies available include rare low-number series, non-European populations, and treatment with methimazole, a drug not used in France and only rarely used in Europe. Our study is one of the first to investigate a mainly carbimazole-induced Grade 4 neutropenia and documented agranulocytosis in Caucasian patients. Physicians must be aware of the most commonly implicated drugs as suspicion is a key element in the diagnosis. Early ATD-induced neutropenia detection may decrease the severity and mortality when the causative drug is withdrawn.

## Figures and Tables

**Table 1 medicines-07-00015-t001:** Clinical features of the study sample at hospital admission.

Clinical Features	n (%)
Sore throat and acute tonsillitis	12 (40%)
Isolated fever (unknown origin)	6 (20%)
Septicemia	4 (13.3%)
Documented pneumonia	2 (6.7%)
Septic shock	2 (6.7%)
Pyelonephritis	1 (3.3%)
Inguinal abscess	1 (3.3%)
Cholecystitis	1 (3.3%)
Cutaneous infection and osteonecrosis	1 (3.3%)

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
