# Peer review of "Severe Neutropenia and Agranulocytosis Related to Antithyroid Drugs: A Study of 30 Cases Managed in A Single Reference Center"

_medicines, 2020, doi:10.3390/medicines7030015_

Round 1

Reviewer 1 Report

The paper by Andrès E. et al. presents data on 30 patients with neutropenia related to antithyroid drugs in Caucasian patients.

The subject of this paper fits the scope of Medicines and the aim of the study is interesting. In my opinion, all procedures have been clearly described, the manuscript is well-written, presented in an intelligible fashion and the language is clear and correct.

Of great importance is the fact that the authors of the paper are practically experts in the field of neutropenia and agranulocytosis, with numerous papers, both original and reviews, published.

They precisely described weak point of their study, although it is obvious that any study performed within such long period of time as 30 years does possess several limitations.

The only my remark concerns the use of benzylthiouracil. Although only two patients developed neutropenia due to the use of benzylthiouracil, it should be stressed in the discussion section that this drug is generally not recommended for the treatment of thyrotoxicosis.

In my opinion this paper is worth publishing in a journal such as Medicines.

Author Response

Dear Reviwer,

Many thanks for your constructive comments.

I have included your comme,nt i  the present version of the manuscript (on benzylthiouracil). Corrections are indicated in the current version in yellow color directly in the text.

Kind regards

Emmanuel Andrès

Reviewer 2 Report

This reviewer would like to thank the authors of this article for their valuable work in the field of drug-induced neutropenia. From my perspective, it is a clear and well written article, although the english still needs some editing. The methodology is appropriate for the objectives of the study. I would suggest including tables/graphs in order to improve the readability of the results section. It is sometimes difficult to keep the attention on the paper with all the data. I will suggest rewritting the conclusions section, as it seems they are logical but no very meaningful in the light of the results.

Line 31: Please consider including a sentence that clearly reflects the impact of your work (the practical implications of it).

Line 48: in Europe and USA

Line 57: The recruiter departments seem to be more a result than a method.

Line 68: This sentence needs some rewritting.

Line 69: I'm not sure about what the CTCAE severity classification ads to the study since neutropenia and agranulocytosis have been previously defined.

Line 123: Charlson comorbidity index has not been included in the methods section. Please, include the reference.

Lines 126, 133: Please, use active ingredients instead of brand names consistently all over the text.

Line 132: I'd suggest deleting this part. Not sure what it adds to the objectives or the discussion of this research.

Line 145: only had

Line 184: I'd strongly recommend including a graph with some of these results in order to reduce the amount of text in this section.

Line 185: were withdrawn

Line 186: patients were immediatly treated

Line 188, 200: Please, use active ingredients instead of brand names consistently all over the text.

Line 201: Do the authors have more information about how the antithyroid therapy was managed after the recovery from the adverse drug reaction? Which antithyroid agent was used after the withdrawal of the causative drug? Since many of these drugs share a common structure (thioureas), this reviewer finds interesting which one could be considered as a safe alternative after the episode.

Line 209: includE

Line 212: unicentric?

Line 330: As previously discussed, the conclussions section need some rework. In my oppinion, they doesn't reflect the value of the work. Please, include more meaningful or practical conclusions (deaths related to this adverse events, specific lessons for the clinical management, drug initiation after the episode, impact of HGF...)

Author Response

Dear Reviewer
Many thanks for your constructive comments
Please find my responses of your comments:
- Line 31: modification of the conclusion
- Line 48: correction done
- Line 57: modification of the phrase
- line 68: correction done
- Line 69: correction done
- Line 123: Charlson's comorbidity index is Added in the section material and method
- Line 126: correction not done. I prefere brand name
- Line 132: correction done
- Line 145: correction done
- Line 184: correction not done. i prefere text
- Line 185 et 186: corrections done
- Line 188 et 200: I prefere brand name (this is the cas in all the literature on this field of médicine)
- Line 201: I have no more information
_ Line 209: correction done
- Line 330: modification of the conclusion
The corrections are indicated in yellow color directly in the text
Kind regards
Emmanuel Andrès

Reviewer 3 Report

The manuscript by Andrès et al is interesting. I have a few comments.

  1. Throughout the manuscript: The number of included individuals is only 30, please do not use any decimal when giving percentage!
  2. Line 57-58: It is called Department of Internal Medicine etc.
  3. Line 132-134: It is strange that only 10% had Grave’s as underlying disease. This is the most common cause of ATD use in most places. Was ATD use not common in Grave’s disease in Strasbourg?
  4. Line 145: Change to had.
  5. Line 153-154: Please add a P-value.
  6. Results: Use median and range if not normally distributed and if normally distributed use mean and standard deviation. Do not use both for each parameter.
  7. Line 177: I guess you mean older than.
  8. Line 186: I think a treated is missing.
  9. I prefer data not shown instead of data not detail. If using the latter please change to data not detailed.
  10. How was the thyrotoxicosis treated when the ATD was ceased? Did they have surgery? RAI?
  11. Calissendorff et al Endocrine Connections 2017 and in a later review in Endocrine the same year describes Lugol treatment in thyrotoxicosis when ATD cannot be use due to adverse events including agranulocytosis. These two articles can be mentioned and cited briefly in the Discussion.
  12. Line 229: I do not understand what you mean by singularly.
  13. Line 244: The sentence is not clear. Please rephrase.
  14. Line 275-277: I do not think it is appropriate to change from one ATD to another if neutropenia/agranulocytosis have occurred. As demonstrated in this paper and correctly stated there are cross-reactions. Patients should have RAI or surgery, often with pretreatment of Lugol solution.
  15. Did the study have ethical approval?

Author Response

Dear Reviwer

Many thanks for your constructive comments

Please find my responses:

  1. No decimal: I prefere decimal. this is the case in all the lierature in the current field of médicine, even in study with small effectif of patient
  2. Line 57 and 58: ok
  3. Line 132-134: carbimazole
  4. Line 145: correction done
  5. Line 153 and line 154: correction done ( p added)
  6. I don't know if it's a normal distribution
  7. Line 177: correction done
  8. Line 186: correction done
  9.  Correction sdone (date not shown)
  10.  Wait and see and surgery
  11. Ok for the references
  12. Line 229: singularly deleted
  13.  Line 244: new phrase
  14.  Line 275-277: correction done
  15. Yes

Round 2

Reviewer 3 Report

The revised manuscript by Andrès et al has improved. I still have a few comments.

  1. When giving your rebuttal, please copy the reviewer’s comments above your response to make it easier to review!
  2. I do not think you have the precision to give one decimal in percentages when you only have 30 individuals. If you had more than 100, yes but not lower numbers. It gives a false sense of precision to give decimals in a case like this. However, when you give means or medians you may have decimals.
  3. Please discuss why only 10% had Grave’s as underlying disease. This is the most common cause of ATD use in most places. Was ATD use not common in Grave’s disease in Strasbourg?
  4. Results: Use median and range if not normally distributed and if normally distributed use mean and standard deviation. Do not use both for each parameter. If you do not know if it is normally distributed or not you have to find this out. Most statistical programs will give you if it is normally distributed or not, if you do not know how to do it, please get some help!
  5. Please mention in the results how the thyrotoxicosis was treated when the ATD was ceased (how many had surgery?), and the longterm outcome of the thyrotoxicosis.

Author Response

Resposes:

1- corrections are indicated in yellow color directly in the text of the new manuscript version; correction are realized for all the section including similarity of more than 50%.

2 - OK. Done.

3 - Carbimazol

4 - OK

5 - Done
